# Chemotherapy activates inflammasomes to cause inflammation-associated bone loss

Chun Wang[1], Khushpreet Kaur[1], Canxin Xu[2], Yousef Abu-Amer[3,4], Gabriel Mbalaviele[1]*

[1]Division of Bone and Mineral Diseases, Washington University School of Medicine, St. Louis, United States; [2]Aclaris Therapeutics, Inc, St. Louis, United States; [3]Department of Orthopaedic Surgery, Washington University School of Medicine, St. Louis, United States; [4]Shriners Hospitals for Children, St. Louis, United States

*For correspondence:
gmbalaviele@wustl.edu

**Abstract** Chemotherapy is a widely used treatment for a variety of solid and hematological malignances. Despite its success in improving the survival rate of cancer patients, chemotherapy causes significant toxicity to multiple organs, including the skeleton, but the underlying mechanisms have yet to be elucidated. Using tumor-free mouse models, which are commonly used to assess direct off-target effects of anti-neoplastic therapies, we found that doxorubicin caused massive bone loss in wild-type mice, a phenotype associated with increased number of osteoclasts, leukopenia, elevated serum levels of danger-associated molecular patterns (DAMPs; e.g. cell-free DNA and ATP) and cytokines (e.g. IL-1β and IL-18). Accordingly, doxorubicin activated the absent in melanoma (AIM2) and NLR family pyrin domain containing 3 (NLRP3) inflammasomes in macrophages and neutrophils, causing inflammatory cell death pyroptosis and NETosis, which correlated with its leukopenic effects. Moreover, the effects of this chemotherapeutic agent on cytokine secretion, cell demise, and bone loss were attenuated to various extent in conditions of AIM2 and/or NLRP3 insufficiency. Thus, we found that inflammasomes are key players in bone loss caused by doxorubicin, a finding that may inspire the development of a tailored adjuvant therapy that preserves the quality of this tissue in patients treated with this class of drugs.

## eLife assessment

This **useful** study, which systematically addresses off-target effects of a commonly used chemotherapy drug on bone and bone marrow cells and which therefore is of potential interest to a broad readership, presents evidence that reducing systemic inflammation induced by doxorubicin limits bone loss to some extent. The demonstration of the effect of systemic inflammation on bone loss is **convincing**. Building on prior work, this study sets the scene for additional genetic and pharmacologic experiments as well as future analyses of the bone phenotypes, which should speak to the mechanisms involved in doxorubicin-induced bone loss – which are not addressed in the current study – and which may substantiate the clinical relevance of targeting inflammation in order to limit the negative impact of chemotherapies on bone quality.

## Introduction

The chemotherapeutic drug, doxorubicin, is widely used for the treatment of breast cancer, bladder cancer, lymphomas, and acute lymphocytic leukemia (*Smith et al., 2010*; *Jang et al., 2023*; *Geffen and Man, 2002*). Despite its success in improving the survival rate of cancer patients, doxorubicin

causes serious adverse effects, including cardiomyopathy, bone marrow suppression, hair loss, and skeletal manifestations (*Cardinale et al., 2015*; *Tacar et al., 2013*; *Coleman et al., 2013*). Bone complications include osteoporosis, a metabolic disease that is characterized by decreased bone mass and deteriorated microarchitecture, and associated with increased risks for the development of late fractures, and morbidities in the elderly populations (*Cauley and Lui, 2009*; *Cawthon et al., 2009*; *Cawthon et al., 2012*). In fact, it was reported that 20–50% of geriatric patients (≥65 years) with a hip fracture die within 1 year of fracture (*Coleman et al., 2013*). Consistent with the dogma that bone resorption by osteoclasts (OCs) and bone formation by osteoblasts is uncoupled in osteoporotic patients, doxorubicin causes bone loss by promoting osteoclastogenesis while suppressing osteoblastogenesis (*Yao et al., 2020*; *Chai et al., 2014*; *Rana et al., 2013*; *Zhou and Kuai, 2020*). Increased production of senescence-associated secretory phenotype, enhanced generation of reactive oxygen species, and dysregulated autophagy and mitochondrial metabolism are proposed mechanisms of doxorubicin-induced bone pathology (*Yao et al., 2020*; *Park et al., 2022*).

Doxorubicin intercalates into DNA thereby impeding the activity of DNA repair enzymes such as topoisomerase II and impairing DNA replication (*Robson et al., 1987*; *Lawrence, 1988*; *Bonner and Lawrence, 1990*; *Abe et al., 2022*; *Tewey et al., 1984*). Defective DNA repair ultimately culminates in genomic instability and cell demise, events that can provoke uncontrollable release of intracellular contents such as DNA and various danger-associated molecular patterns (DAMPs). DNA-enriched entities include neutrophil extracellular traps (NETs), web-like structures in which DNA is decorated with peptides, some of which have anti-microbial and inflammatory properties (*Komada et al., 2018*; *Apel et al., 2021*). NETs can also propagate inflammation following their engulfment by phagocytes (*Boccia et al., 2022*; *Blayney and Schwartzberg, 2022*; *Nakazawa et al., 2016*; *Jeong et al., 2021*). Since DNA normally resides in the nucleus and mitochondria, its presence in the cytoplasm is detected by DNA sensors, including absent in melanoma 2 (AIM2), and cyclic guanosine monophosphate-adenosine monophosphate synthase, which can trigger immune responses aimed at eliminating the mislocated DNA (*Sun et al., 2013*; *Zhang et al., 2014*; *Rathinam et al., 2010*). Oxidized DNA and various DAMPs can also be sensed by NLRP3 (*Shimada et al., 2012*; *Lu et al., 2014*; *Xian et al., 2022*). Upon recognition of DAMPs or pathogen-associated molecular patterns (PAMPs), AIM2 and NLRP3 assemble protein platforms comprising the adaptor protein apoptosis-associated speck-like protein containing a CARD (ASC) and caspase-1. These protein complexes known as inflammasomes are responsible for the maturation of pro-interleukin-1β (pro-IL-1β) and pro-IL-18 to IL-1β and IL-18, respectively (*Guo et al., 2015*; *Sharma and Kanneganti, 2021*; *Wang et al., 2021*). Inflammasome-comprising caspase-1 also cleaves gasdermin D (GSDMD), generating N-terminal fragments, which form IL-1β- and IL-18-secreting conduits, and cause the inflammatory cell death, pyroptosis (*Guo et al., 2015*; *Sharma and Kanneganti, 2021*; *Wang et al., 2021*). While acute activation of inflammasomes is important for the clearance of the perceived danger and restoration of homeostasis, chronic or excessive stimulation of these safeguard mechanisms can cause disease.

Inflammasomes are involved in skeletal pathophysiology. Gain-of-function mutations of *NLRP3* cause skeletal abnormalities in humans (*Aksentijevich et al., 2002*; *Feldmann et al., 2002*), which are reproduced to a great extent in knock-in mice expressing NLRP3 harboring mutations found in these patients (*Bonar et al., 2012*; *Qu et al., 2015*; *Snouwaert et al., 2016*; *Wang et al., 2017*). In normal mice, degraded bone matrix components, which are released during bone resorption, promote inflammasome activation and OC differentiation (*Alippe et al., 2017*). Age-associated bone loss has also been linked to chronic low-grade inflammation mediated by the NLRP3 inflammasome (*Youm et al., 2013*). More relevant to this study, radiation, which is also used as an anti-neoplastic therapy, causes bone loss through GSDMD downstream of AIM2 and NLRP3 inflammasomes, but not NLR family caspase recruitment domain containing protein 4 (NLRC4) inflammasome (*Xiao et al., 2020*). Collectively, the bone phenotypes of genetically or pharmacologically activated inflammasome sensors suggest that the fate of bone cells can be influenced by inflammation driven by inflammasomes, which are mainly activated in myeloid cells (*Brydges et al., 2009*; *Meng et al., 2009*). This view provides a strong rationale for exploring the role of inflammasome pathways in bone loss induced by off-target actions of doxorubicin as this agent causes the death of cancer and bystander normal cells, thereby releasing DAMPs such as ATP and DNA, which activate these pathways.

We used non-tumor-bearing mouse models, which are commonly used to assess off-target outcomes of anti-neoplastic therapies (*Park et al., 2022*; *Borniger et al., 2015*; *Harrison et al.,*

*1980*; *Yao et al., 2020*) to study bone adverse effects of doxorubicin. We found that doxorubicin caused massive bone loss in wild-type (WT) mice, a phenotype associated with increased number of OCs, leukopenia, and cytokinemia. These outcomes implicated the AIM2 and NLRP3 inflammasomes as they were attenuated upon genetic inactivation of these sensors. Thus, our results show that inflammasomes are key players in bone loss caused by doxorubicin, a finding that may enable the implementation of novel strategies for chemotherapy-related bone complications.

## Results

### Doxorubicin causes bone loss

To determine the effects of doxorubicin on bone mass, femurs of 10-week-old WT female mice were analyzed by VivaCT before (baseline) and 4 weeks after a single intraperitoneal injection of 5 mg/kg doxorubicin or vehicle. Doxorubicin, but not the vehicle, caused bone loss (*Figure 1—figure supplement 1A–D*). Doxorubicin also caused bone loss in 10-week-old WT male mice, a response that was associated with increased OC number and surface (*Figure 1A–E*) and decreased bone formation (*Figure 1—figure supplement 1E*). These findings were consistent with the recently reported stimulatory and suppressive effects of this drug on bone resorption and formation in WT female mice, respectively (*Yao et al., 2020*). Since doxorubicin inflicted bone damage independently of the sex, afterward mechanistic studies focused mainly on male mice and revolved around innate immune responses, which regulate OC-mediated bone resorption in pathological conditions.

### Doxorubicin causes cytokinemia, leukopenia, release of DAMPs, and NETosis in vivo

Doxorubicin induces inflammatory responses in patients and experimental models (*Wittenburg et al., 2019*). Accordingly, WT mice exposed to doxorubicin for 3 days exhibited higher serum levels of IL-1β, IL-18, IL-6, and TNF-α compared to vehicle-injected counterparts (*Figure 2A–D*). Levels of these inflammatory cytokines inversely correlated with the abundance of white blood cells (WBCs; *Figure 2E*). While doxorubicin lowered the number of circulating lymphocytes and monocytes, the number of neutrophils, the most abundant immune cells in the blood, increased 2 hr post-drug exposure before progressively returning to baseline levels. Consistent with the leukopenic outcome, levels of ATP, which is released by dead cells, were higher in doxorubicin-treated mice compared to vehicle-treated cohorts (*Figure 2F*). To gain further insight into the mechanisms of leukopenia, we focused on neutrophils, the most abundant leukocytes in blood. These cells exhibit morphological changes such as NET extrusion upon exposure to PAMPs or sterile DAMPs, and eventually undergo NETosis (*Komada et al., 2018*; *Apel et al., 2021*). To determine the effects of doxorubicin on NET formation, we measured serum levels of NET components in mice treated with vehicle or this drug for 48 hr. We found that doxorubicin induced NET formation as evidenced by increased levels of citrullinated histone 3 (Cit-H3; *Figure 2G*), myeloperoxidase (MPO; *Figure 2H*), and cell-free DNA (cfDNA) (*Figure 2I and J*). Thus, doxorubicin causes cytokinemia, a response that is associated with increased cell death and decreased number of WBCs.

### Doxorubicin activates inflammasome-dependent and -independent pathways, and causes macrophage pyroptosis

The effects of doxorubicin on macrophages have been reported (*Chen et al., 2023*; *Saleh et al., 2021*). To test the hypothesis that these cells were implicated in the inflammatory phenotype of mice treated with doxorubicin, we treated bone marrow-derived macrophages (BMMs) with lipopolysaccharide (LPS) for 3 hr to induce the expression of inflammasome components (*Wang et al., 2021*), then with various concentrations of this drug for 16 hr. Within its reported potent concentrations (1.5–12 μM) (*Saleh et al., 2021*; *Eljack et al., 2022*), doxorubicin did not induce IL-1β secretion, but it significantly caused the release of lactate dehydrogenase (LDH; *Figure 3A and B*), a marker of cell death as it is released only upon plasma membrane rupture (*Wang et al., 2021*). Since LDH release was not induced by doxorubicin in a dose-dependent manner, this response may be the result of non-selective cytotoxic actions of this drug. By contrast, doxorubicin promoted IL-1β and LDH release by LPS-primed BMMs in a dose-dependent fashion, with the maximal effect on IL-1β secretion achieved at 6 μM (*Figure 3A*). Unexpectedly, LPS attenuated LDH release induced by low doxorubicin

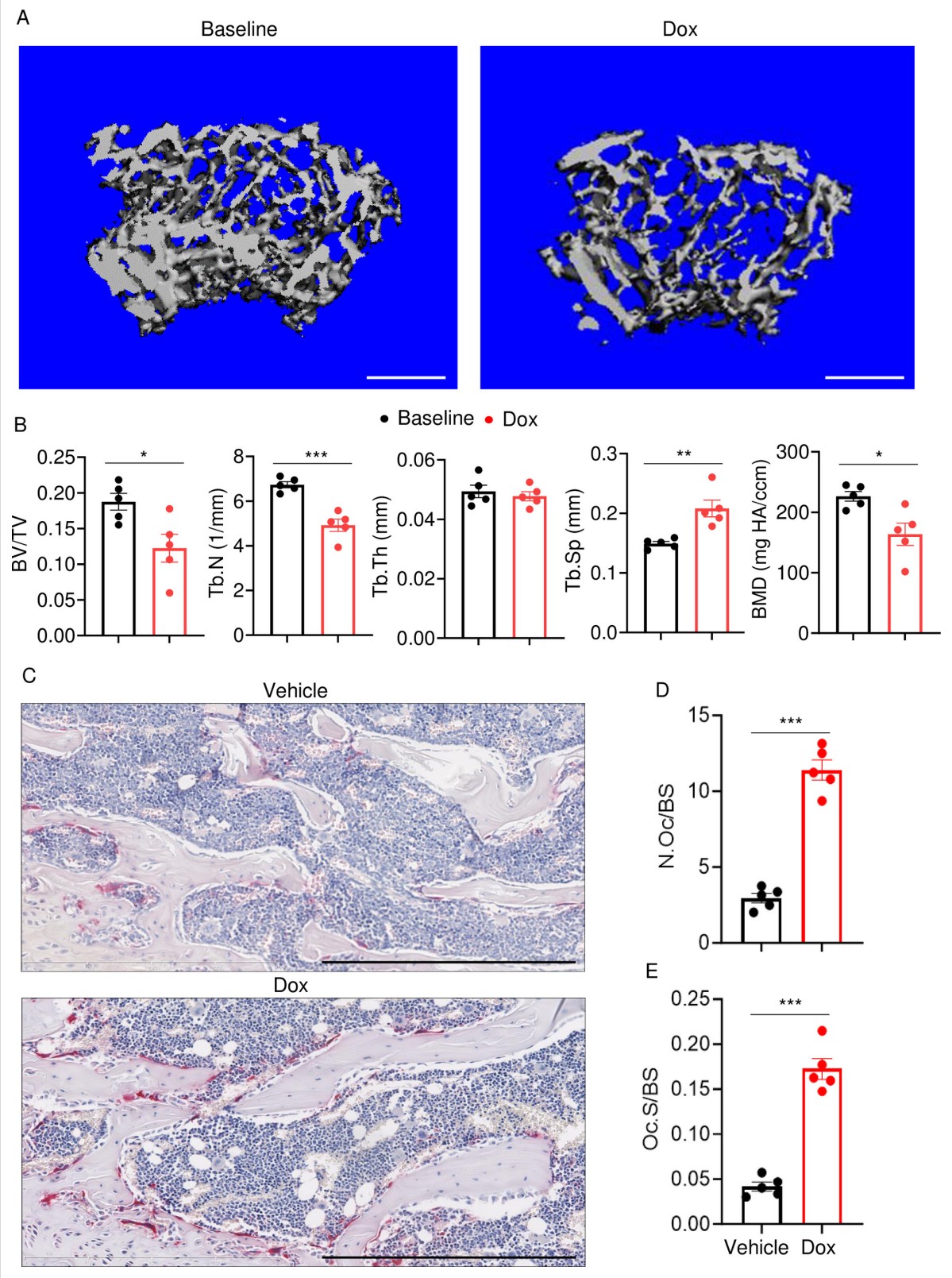

**Figure 1.** Doxorubicin causes bone loss in male mice. Femurs from WT male mice were analyzed by VivaCT before (baseline) and 4 weeks after a single intraperitoneal injection of 5 mg/kg doxorubicin. Cross sections of 3D reconstructions. Scale bars: 500 μm (**A**), bone parameters (**B**), and femurs from WT male mice (**C–E**) were analyzed 4 weeks after a single intraperitoneal injection of vehicle or doxorubicin. Specimens were stained for tartrate-resistant acidic phosphatase (TRAP) activity. Representative images. Scale bars: 500 μm (**C**), N.Oc/BS (**D**), Oc.S/BS (**E**). N=5 mice/group. Data are mean ± SEM.

*Figure 1 continued on next page*

*Figure 1 continued*

Student's t-test was used. *p<0.05; **p<0.01; ***p<0.001. BMD, bone mineral density; BV/TV, bone volume/total volume; Dox, doxorubicin; N.Oc/BS, OC number/bone surface; Oc.S/BS, OC surface/bone surface; OC, osteoclast; ns, not significant; Tb.N, trabecular number; Tb.Th, trabecular thickness; Tb.Sp, trabecular separation; WT, wild-type.

The online version of this article includes the following source data and figure supplement(s) for figure 1:

**Source data 1.** Micro-computed tomography (μCT) analysis in *Figure 1B*.

**Source data 2.** Tartrate-resistant acidic phosphatase (TRAP) staining analysis in *Figure 1D and E*.

**Figure supplement 1.** Doxorubicin causes bone loss in female mice.

**Figure supplement 1—source data 1.** Micro-computed tomography (μCT) analysis in *Figure 1—figure supplement 1C–D*.

**Figure supplement 1—source data 2.** Dynamic bone parameters analysis in *Figure 1—figure supplement 1E*.

---

concentrations (1. 5 and 3 µM) (*Figure 3B*). To further gain insight into the mechanism of action of doxorubicin, we measured the levels of ATP, which is released by dead cells and activates multiple pathways, including the NLRP3 inflammasome (*Karmakar et al., 2016*; *Carta et al., 2015*). ATP levels were higher in the supernatants of doxorubicin-treated BMMs compared to controls, a response that was further enhanced by LPS (*Figure 3C*). Collectively, these results suggest that BMMs underwent pyroptosis in the presence of LPS and doxorubicin, releasing DAMPs such as ATP.

To reinforce the proposition that doxorubicin activates inflammasomes, we assessed the effects of this drug on the formation of ASC specks, a readout of inflammasome-activated states (*SAtutza et al., 2013*). As anticipated, LPS induced ASC speck formation only in the presence of nigericin, a well-known trigger of NLRP3 inflammasome assembly signals (*Figure 3D, E, and H*). Likewise, doxorubicin induced ASC speck formation only in LPS-primed BMMs (*Figure 3F, G, and H*). Next, we performed immunoblotting to analyze the expression of NLRP3 and AIM2 since these sensors assemble inflammasomes in response to DAMPs such as ATP and DNA, which were released by doxorubicin-damaged cells. We also determined the expression of other key components of these pathways such as caspase-1 and gasdermins. LPS induced the expression of NLRP3, but not AIM2, caspase-1, caspase-3, GSDMD, and GSDME (*Figure 4*). Levels of caspase-1 (p10) and GSDMD (p30) fragments, which are generated upon inflammasome activation, were higher in cells treated with LPS+doxorubicin compared to doxorubicin alone (*Figure 4*, *Figure 4—figure supplement 2*). GSDMD (p10) and GSDME (p35) fragments, which are proteolytically generated by caspase-3 (*Figure 4—figure supplement 1A and B*), were also detected, but their abundance was comparable between cells exposed to LPS+doxorubicin and doxorubicin alone (*Figure 4* and *Figure 4—figure supplement 2*). Together, these results suggest that doxorubicin activates both caspase-1 and caspase-3, which cleave GSDMD and GSDME, ultimately, causing pyroptosis and IL-1β release.

## Doxorubicin activates inflammasome-dependent and -independent pathways, and causes NETosis in vitro

To further support the conclusion that doxorubicin induced the formation of NETs, first, we analyzed the expression of some key players directly or indirectly involved in this process. Incubation of mouse bone marrow neutrophils with LPS resulted in increased NLRP3 expression (*Figure 5A*). The abundance of caspase-1 (p10) and GSDMD (p30) was indistinguishable between cells exposed to LPS+doxorubicin and doxorubicin alone, likely as the result of cell death as the levels of β-actin used as loading control were markedly reduced in samples from these cells (*Figure 5A* and *Figure 5—figure supplement 1*). Levels of GSDMD (p10) and GSDME (p35) fragments were also similar between LPS+doxorubicin compared to doxorubicin alone (*Figure 5A*, *Figure 4—figure supplement 1*, *Figure 5—figure supplement 1*). LPS also induced IL-1β secretion, a response that was enhanced by doxorubicin in a dose-dependent manner (*Figure 5B*). LPS reduced baseline as well as doxorubicin-induced LDH release (*Figure 5C*). In sum, unchallenged neutrophils released LDH, a response that aligned with the presence of functional caspase-3, GSDMD, and GSDME fragments in these cells. However, LPS was required for optimal NLRP3 expression, caspase-1 activation, and IL-1β production by neutrophils. Although neutrophils secreted higher levels of IL-1β in response to doxorubicin, they were highly sensitive to the cytotoxic effects of this drug.

Next, we performed immunofluorescence to visualize NET components in neutrophils treated with vehicle or this drug for 16 hr. Consistent with in vivo results, Cit-H3 and MPO were detected in cells

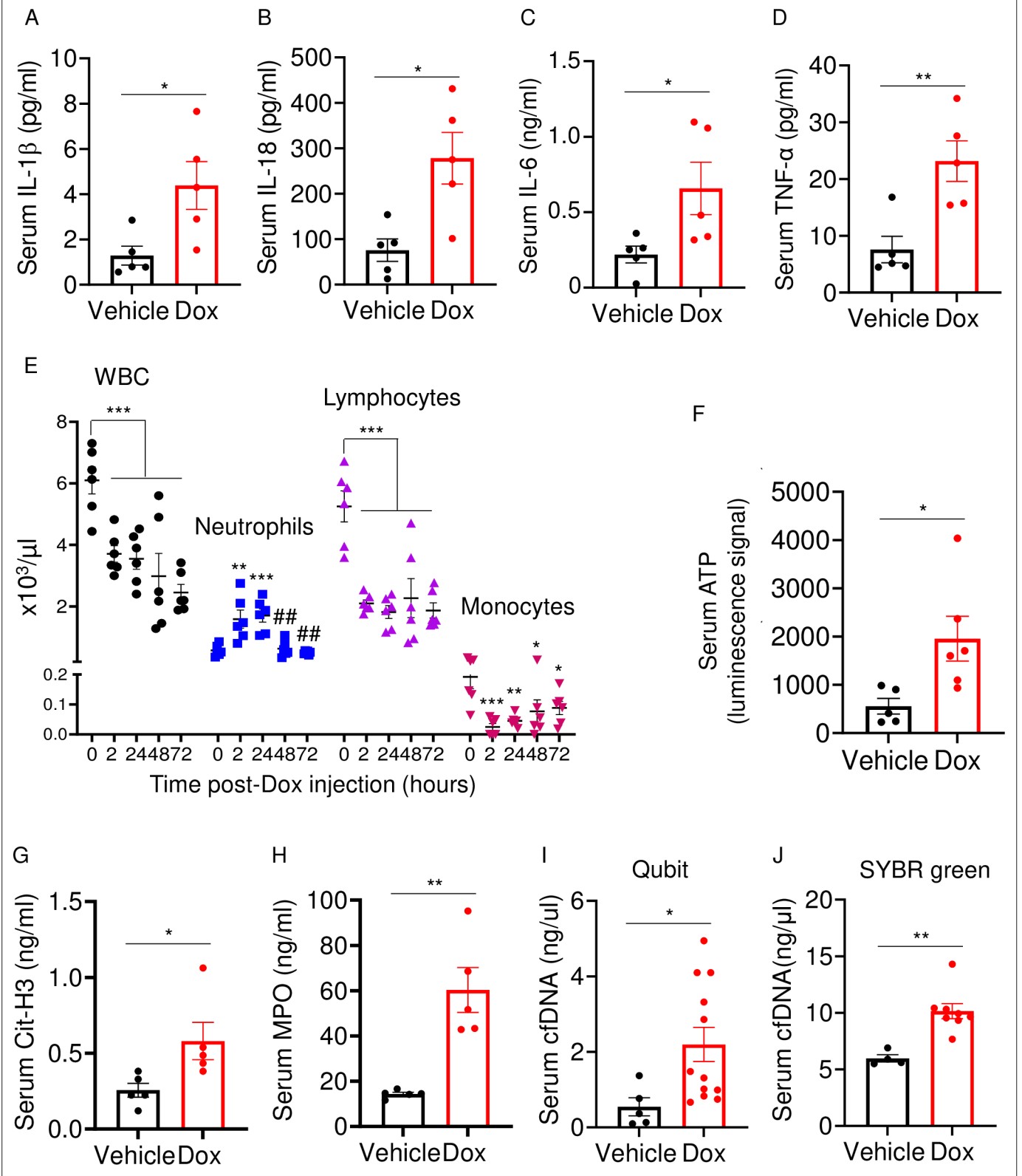

**Figure 2.** Doxorubicin causes cytokinemia, leukopenia, release of danger-associated molecular patterns (DAMPs), and NETosis in vivo. Twelve-week-old WT mice were exposed to a single dose of vehicle or 5 mg/kg doxorubicin. Serum samples were harvested 3 days (**A–D**) or 2 days (**F–J**) later and analyzed by MSD (IL-1β, IL-6, and TNF-α) or ELISA (Cit-H3, IL-18, and MPO). Blood was collected for cell counts at the indicated time-points after a single dose of 5 mg/kg doxorubicin injection (**E**). cfDNA was measured using Qubit (**I**) or SYBR green (**J**). Data are mean ± SEM. N=5–12 mice/group.

*Figure 2 continued on next page*

*Figure 2 continued*

*p<0.05; **p<0.01; ***p<0.001 vs 0 hr; ##p<0.01 vs 2 or 24 hr. Student's t-test (**A–D, F–J**) and one-way ANOVA (**E**) were used. cfDNA, cell-free DNA; Cit-H3, citrullinated histone H3; Dox, doxorubicin; IL, interleukin; MPO, myeloperoxidase; WBCs, white blood cells.

The online version of this article includes the following source data for figure 2:

**Source data 1.** Serum samples analysis of inflammatory cytokines, ATP, citrullinated histone 3 (Cit-H3), myeloperoxidase (MPO), and cfDNA in *Figure 2A–D and F–J*.

**Source data 2.** Complete blood count in *Figure 2E*.

treated with doxorubicin but not with vehicle (*Figure 6A*). Accordingly, levels of cfDNA were higher in doxorubicin-exposed cultures compared to untreated or cultures treated with LPS (*Figure 6B*). To determine the biological relevance of cfDNA while modeling the in vivo bone microenvironment, we assessed the impact of degrading cfDNA with DNase I on IL-1β release by cultured whole bone marrow cells, which comprised various cell types, including neutrophils and macrophages. IL-1β levels were higher in cells treated with LPS and doxorubicin compared to LPS, an outcome that was inhibited by DNase I (*Figure 6C*). Thus, doxorubicin promoted IL-1β release, a response that correlated with its effects on NET formation and abundance of cfDNA.

## AIM2 and NLRP3 inflammasomes are involved in bone-damaging effects of doxorubicin

We hypothesized that doxorubicin induced IL-1β and IL-18 release by activating the AIM2 and NLRP3 inflammasomes as blood levels of their activators (DNA, ATP) were increased in response to doxorubicin administration (*Figure 2F, I, J*). To test this idea, we measured the effects of this drug on IL-1β and LDH release by WT, *Aim2*-/-, *Nlrp3*-/-, *Aim2*-/-; *Nlrp3*-/-, or *Casp1*-/- BMMs and neutrophils. LPS induction of IL-1β and LDH release by BMMs required doxorubicin, a response that was significantly reduced in *Aim2*-/- or *Nlrp3*-/- cells (*Figure 7A and B*). LPS induced IL-1β secretion by neutrophils, an outcome that was enhanced by doxorubicin, and comparably attenuated in all mutants (*Figure 7C and D*). Differences in LDH release were marginal, perhaps because baseline levels of this readout were high, consistent with the short lifespan of these cells in vitro. Because inflammation leads to bone loss, we assessed bone outcomes of 10-week-old WT, *Aim2*-/- and/or *Nlpr3*-/-, 4 weeks after receiving a single dose of 5 mg/kg doxorubicin or vehicle. Administration of doxorubicin to WT mice caused bone loss associated with increased OC number and surface (*Figure 7E–I* and *Figure 7— figure supplement 1A–C*), consistent with the results shown above (*Figure 1*). These responses were reduced slightly in *Nlrp3*-deficient mice, but significantly in *Aim2* null mice. *Aim2*-/- male mice lost bone comparably to *Aim2*-/-;*Nlrp3*-/- and *casp1*-/- counterparts (*Figure 7E–G*). Similar trends in bone changes were observed in female mice, though *Aim2*-/- mice were more osteopenic than *casp1*-/- mice (*Figure 7—figure supplement 2A–D*). Collectively, our results suggest that the AIM2 and NLRP3 inflammasomes participate to various extent in doxorubicin bone-damaging effects. They also suggest that inflammasome-independent actions of this drug on bone are not negligible.

## Discussion

We found that the AIM2 inflammasome and the NLRP3 inflammasome to a lesser extent played an important role in bone-damaging effects of doxorubicin. The comparable bone phenotype of *Aim2*-/-;*Nlrp3*-/- and *casp1*-/- mice suggested that the AIM2 and NLRP3 inflammasomes were the primary mediators of doxorubicin actions. Because doxorubicin activates several pathways, some of which interact or overlap with inflammasome functions (e.g. senescence factors), the remaining bone loss in compound mutant mice was expected. The interplay among these pathways may have accounted for the sex differences in the bone outcomes of inflammasome insufficiency as residual doxorubicin-induced bone loss was higher in *Aim2*-/- and *Aim2*-/-;*Nlrp3*-/- female mice than in male counterparts. Sexual dimorphic actions of inflammasomes were not unprecedented as uneven severity of athero-sclerosis was found in male and female *Nlrp3*-deficient mice with gonadal insufficiency (*Chen et al., 2020*) and sex-dependent differential activation of AIM2 and NLRP3 inflammasomes in macrophages from systemic lupus erythematous had been reported (*Yang et al., 2015*). Although the impacts of doxorubicin on bone pathology are complex, including its direct actions on bone cells (*Yao et al.,*

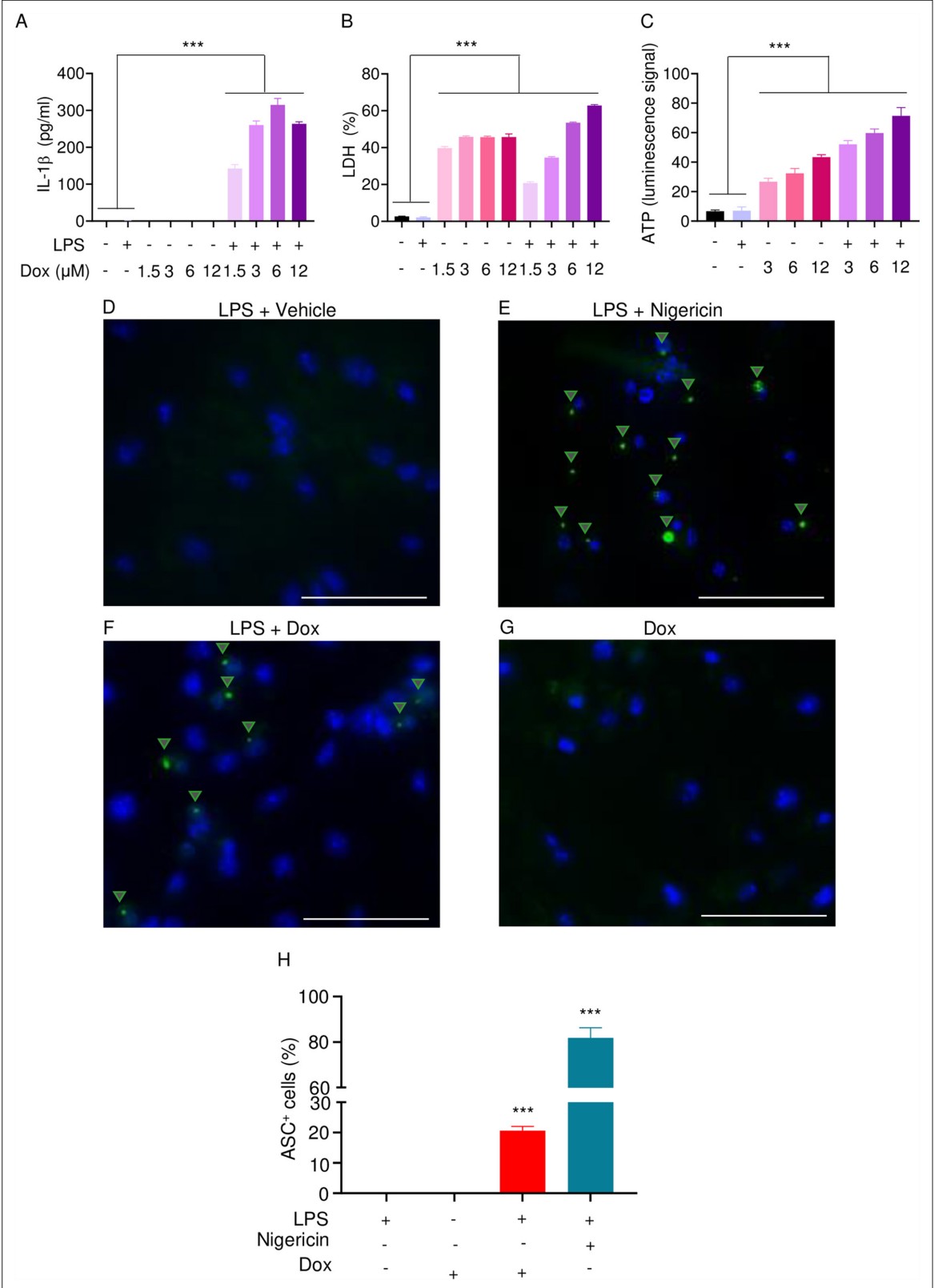

**Figure 3.** Doxorubicin activates inflammasomes and causes macrophage pyroptosis. WT bone marrow-derived macrophages (BMMs) were left untreated or primed with LPS for 3 hr, then treated with various doxorubicin concentrations for 16 hr. IL-1β (**A**), LDH (**B**), and ATP (**C**) in the conditioned media were measured by enzyme linked immunosorbent assay (ELISA), the cytotoxicity detection kit, or ATP detection kit, respectively. WT BMMs from ASC-citrine mice were primed with 100 ng/ml LPS for 3 hr and treated or not with 15 μM nigericin for 30 min or 10 μM doxorubicin for 16 hr (**D–F**). Non-

*Figure 3 continued on next page*

*Figure 3 continued*

primed cells were also treated with 10 μM doxorubicin for 16 hr (**G**). Scale bars: 50 μm. ASC specks were visualized under fluorescence microscopy and quantified using ImageJ. Quantitative data (**H**). Data are mean ± SEM from experimental triplicates and represent at least two independent experiments. ***p<0.001 vs. untreated- or LPS-treated cultures. One-way ANOVA. ASC, apoptosis-associated speck-like protein containing a CARD; ATP, adenosine triphosphate; Dox, doxorubicin; IL-1β, interleukin-1β; LDH, lactate dehydrogenase; LPS, lipopolysaccharide; WT, wild-type.

The online version of this article includes the following source data for figure 3:

**Source data 1.** IL-1β, lactate dehydrogenase (LDH), and ATP analysis of bone marrow-derived macrophages (BMMs) supernatant in *Figure 3A–C*.

**Source data 2.** ASC⁺ cells analysis in *Figure 3H*.

---

*2020*; *Chai et al., 2014*; *Rana et al., 2013*; *Zhou and Kuai, 2020*), our investigation focused on immune cells, and found that the inactivation of inflammasomes is sufficient to attenuate the drug's bone-damaging effects.

Doxorubicin causes neutropenia, lymphopenia, and anemia in patients (*Gatti et al., 2018*; *Boccia et al., 2022*). Consistent with the clinical situation, administration of doxorubicin to mice caused leukopenia, which correlated with lymphocytopenia and monocytopenia, and was associated with fluctuations in neutrophil counts. Since neutrophils were highly sensitive to the cytotoxic effects of doxorubicin, the transient neutrophilic effects of this drug in mice may be the result of emergency granulopoiesis, a physiological response that is rapidly triggered to restore adequate neutrophil number. This leukopenic outcome was consistent with increased serum levels of cell death-associated DAMPS (ATP and cfDNA) and our results showing that doxorubicin activated the effectors of apoptosis, pyroptosis, and NETosis, including caspase-1, caspase-3, GSDMD, and GSDME. Since pyroptosis and NETosis promote inflammation and immune responses, we argued that they accounted for the cytokinemic effects of doxorubicin. Other studies, however, found that doxorubicin reduced NET formation in cancer models and by human neutrophils in vitro (*Lu et al., 2021*; *Khan et al., 2019*). This discrepancy may be due to differences in the experimental models and cell context-dependent actions of doxorubicin. Other limitations of our study include its focus on: (i) macrophages and neutrophils while oversighting lymphocytes or even other cells bone microenvironment such as mesenchymal and adipocytes whose fate is affected by this drug (*Fan et al., 2018*; *Fan et al., 2017*; *Wang et al., 2012*; *Buttiglieri et al., 2011*), which were also targeted by doxorubicin; (ii) immune cells without assessing the direct effects of doxorubicin on bone cells, as noted above; and (iii) the use of the tumor-free model as immune responses can differ significantly in the absence or presence of cancer cells. Despite these limitations, our findings point to a novel mechanism of action for doxorubicin in bone.

DNA accumulates in the cytoplasm as the result of genomic instability, damaged mitochondria, or lysed intracellular pathogens. Extracellular DNA from pathogens, NETotic, or pyroptotic cells can be internalized and culminate in the cytoplasm. In either case, sensors such as AIM2 detect mislocated DNA in the cytoplasm and trigger inflammatory responses (*Boccia et al., 2022*; *Blayney and Schwartzberg, 2022*; *Nakazawa et al., 2016*; *Jeong et al., 2021*). This view was consistent with our data showing a correlation between the levels of extracellular DNA and IL-1β as well as by the inhibition of IL-1β secretion by DNase I. We also found that doxorubicin activated the NLRP3 inflammasome and induced the release of ATP, a well-known activator of the NLRP3 inflammasome. Whether doxorubicin activated the NLRP3 inflammasome directly by perturbing the plasma membrane or indirectly via generation of secondary signals such as ATP remained unclear.

By showing that inflammasomes are key players in bone loss caused by doxorubicin, this work advances our knowledge on potential mechanisms of action of this drug on this tissue (*Figure 8*). This insight is difficult to get in clinical situations because this chemotherapeutic is employed not alone but in conjunction with other medications (*Zimny, 1988*; *Svendsen et al., 2017*; *Müller et al., 2020*).

## Materials and methods

### Animals

WT, R26-CAG-ASC-citrine (030744), *Aim2⁻/⁻* (013144), and *Nlrp3⁻/⁻* (021302) mice were purchased from The Jackson Laboratory (Sacramento, CA, USA). *Casp1⁻/⁻* were kindly provided by Dr. Thirumala-Devi Kanneganti (St. Jude Children's Research Hospital). *Aim2⁻/⁻* mice and *Nlrp3⁻/⁻* mice were intercrossed to generate *Aim2⁻/⁻;Nlrp3⁻/⁻* mice. All mice were on the C57BL/6J background and mouse genotyping was

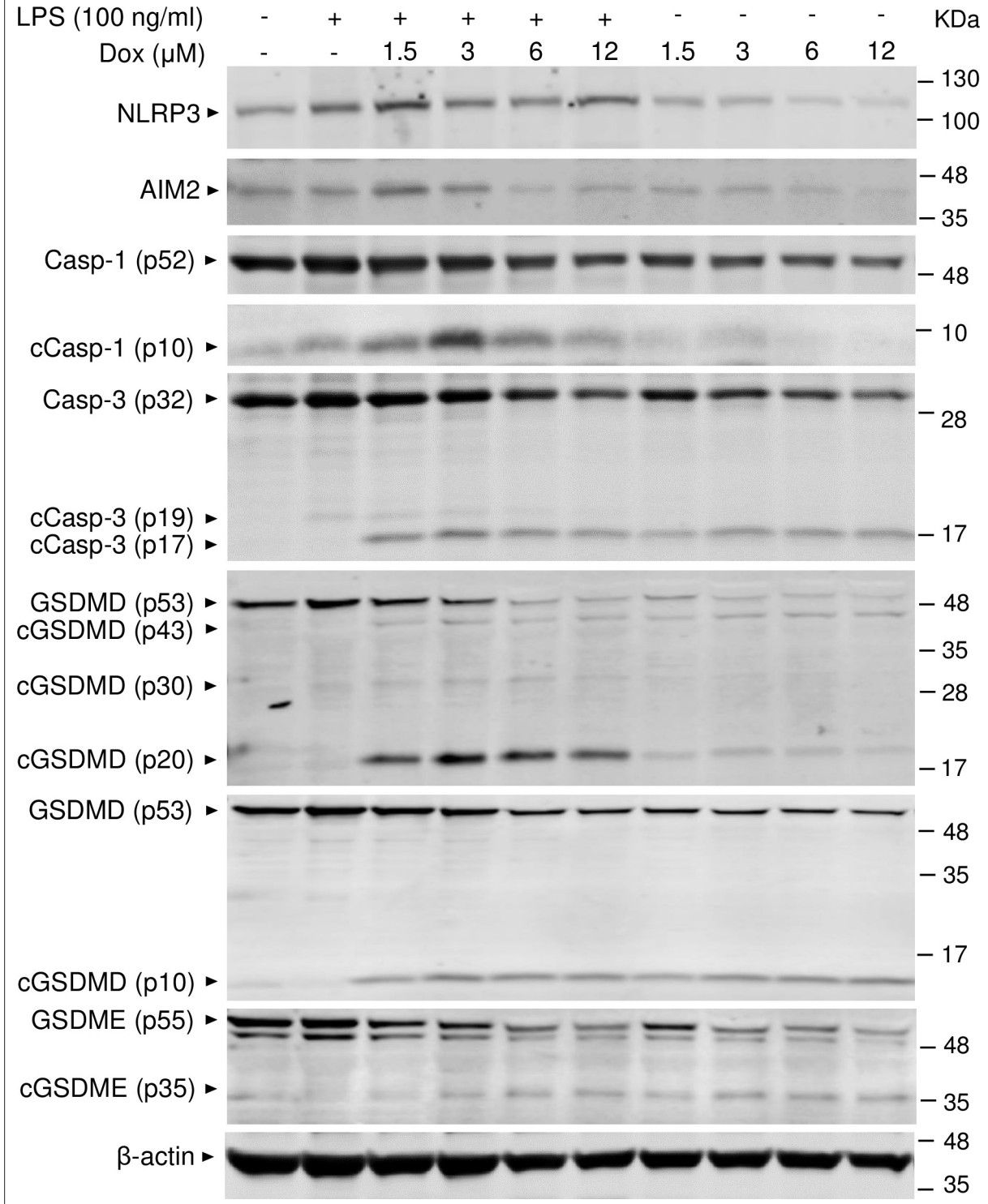

**Figure 4.** Doxorubicin activates inflammasome-dependent and -independent pathways in macrophages. WT bone marrow-derived macrophages (BMMs) were left untreated or primed with LPS for 3 hr, then treated with various doxorubicin concentrations for 16 hr. Whole cell lysates were analyzed by immunoblotting. Data are representative of at least three independent experiments. AIM2, absent in melanoma 2; cCasp, cleaved caspase; cGSDM, cleaved gasdermin; LPS, lipopolysaccharide; Dox, doxorubicin; WT, wild-type.

The online version of this article includes the following source data and figure supplement(s) for figure 4:

**Source data 1.** Original file for the western blot analysis in *Figure 4* (NLRP3).

**Source data 2.** Original file for the western blot analysis in *Figure 4* (AIM2).

*Figure 4 continued on next page*

*Figure 4 continued*

**Source data 3.** Original file for the western blot analysis in *Figure 4* (caspase-1).

**Source data 4.** Original file for the western blot analysis in *Figure 4* (cleaved caspase-1).

**Source data 5.** Original file for the western blot analysis in *Figure 4* (caspase-3/cleaved caspase-3).

**Source data 6.** Original file for the western blot analysis in *Figure 4* (gasdermin D [GSDMD]/cleaved GSDMD).

**Source data 7.** Original file for the western blot analysis in *Figure 4* (gasdermin D [GSDMD]/cleaved GSDMD [(p10)]).

**Source data 8.** Original file for the western blot analysis in *Figure 4* (GSDME/cleaved GSDME).

**Source data 9.** Original file for the western blot analysis in *Figure 4* (β-actin).

**Source data 10.** Original images of the relevant western blot analysis (NLRP3, AIM2, caspase-1/cleaved caspase-1, caspase-3/cleaved caspase-3, gasdermin D [GSDMD]/cleaved GSDMD, GSDME/cleaved GSDME, and β-actin) with highlighted bands and sample labels in *Figure 4*.

**Figure supplement 1.** Cleavage sites of gasdermin D (GSDMD) and GSDME.

**Figure supplement 2.** Doxorubicin activates inflammasome-dependent and -independent pathways in macrophages.

**Figure supplement 2—source data 1.** Western blot analysis of bone marrow-derived macrophages (BMMs) in *Figure 4—figure supplement 2*.

performed by PCR. All procedures were approved by the Institutional Animal Care and Use Committee (IACUC) of Washington University School of Medicine in St. Louis. All experiments were performed in accordance with the relevant guidelines and regulations described in the IACUC-approved protocol 22-0335.

## Doxorubicin administration and VivaCT analysis

The femurs of 10-week-old mice were analyzed by VivaCT 2 weeks before (baseline) and 4 weeks after a single intraperitoneal injection (i.p.) of 5 mg/kg doxorubicin (Sigma-Aldrich, MO, USA) formulated in $H_2O$ at 1 mg/ml or vehicle. For bone analysis, mice were anesthetized with isoflurane and trabecular volume in the distal femoral metaphysis of the right leg was measured using VivaCT 40 (Scanco Medical AG, Zurich, Switzerland) set at 70 kVp, 114 μA, and 20 μm resolution as previously described (*Yao et al., 2020*). For the trabecular bone compartment, contours were traced on the inside of the cortical shell using 2D images of the femoral metaphysis. The end of the growth plate region was used as a landmark to establish a consistent location for starting analysis, and the next 50 slices were analyzed. The following trabecular parameters are reported for all VivaCT experiments: bone volume over total volume, trabecular number, trabecular thickness, trabecular separation, and volumetric bone mineral density.

## Histomorphometry

For static histomorphometry, the femurs were fixed in 10% neutral buffered formalin overnight and decalcified in 14% (wt/vol) EDTA, pH 7.2, for 10–14 days at room temperature. Fixed femurs were embedded in paraffin, sectioned at 5 μm thicknesses, and mounted on glass slides. The sections were stained with tartrate-resistant acidic phosphatase as described previously (*Wang et al., 2018*). For dynamic histomorphometry, mice were i.p. injected with 10 mg/kg calcein green (Sigma-Aldrich, MO, USA) and 6 days later with 50 mg/kg alizarin red (Sigma-Aldrich, MO, USA). Mice were euthanized 2 days after the second injection. The tibias were collected and fixed in 10% neutral buffered formalin overnight, embedded in methyl methacrylate, and sectioned at 7–10 μm. Images were taken using a Nanozoomer 2.0 HT whole slide scanner (Hamamatsu Photonics, Shizuoka, Japan) at ×20 magnification. Bioquant Osteo software (v18.2.6; Bioquant Image Analysis Corp, TN, USA) was used for image analysis. Measurements of dynamic bone histomorphometry were calculated from fluorochrome double labels at the endocortical surfaces as previously described (*Xiao et al., 2020*).

## Serum assays

Blood was collected by cardiac puncture and was allowed to clot at room temperature. Serum obtained after centrifugation at 2000×*g* for 10 min was used for various assays. Cytokine and chemokine levels were measured by V-PLEX Plus Proinflammatory Panel 1 Mouse Kit (Meso Scale Diagnostics, MD USA), except IL-18, which was analyzed by enzyme linked immunosorbent assay (ELISA) kit (Sigma-Aldrich, MO, USA). The levels of Cit-H3 and MPO were determined by ELISA kits (Abcam, MA, USA, and Cayman, MI, USA).

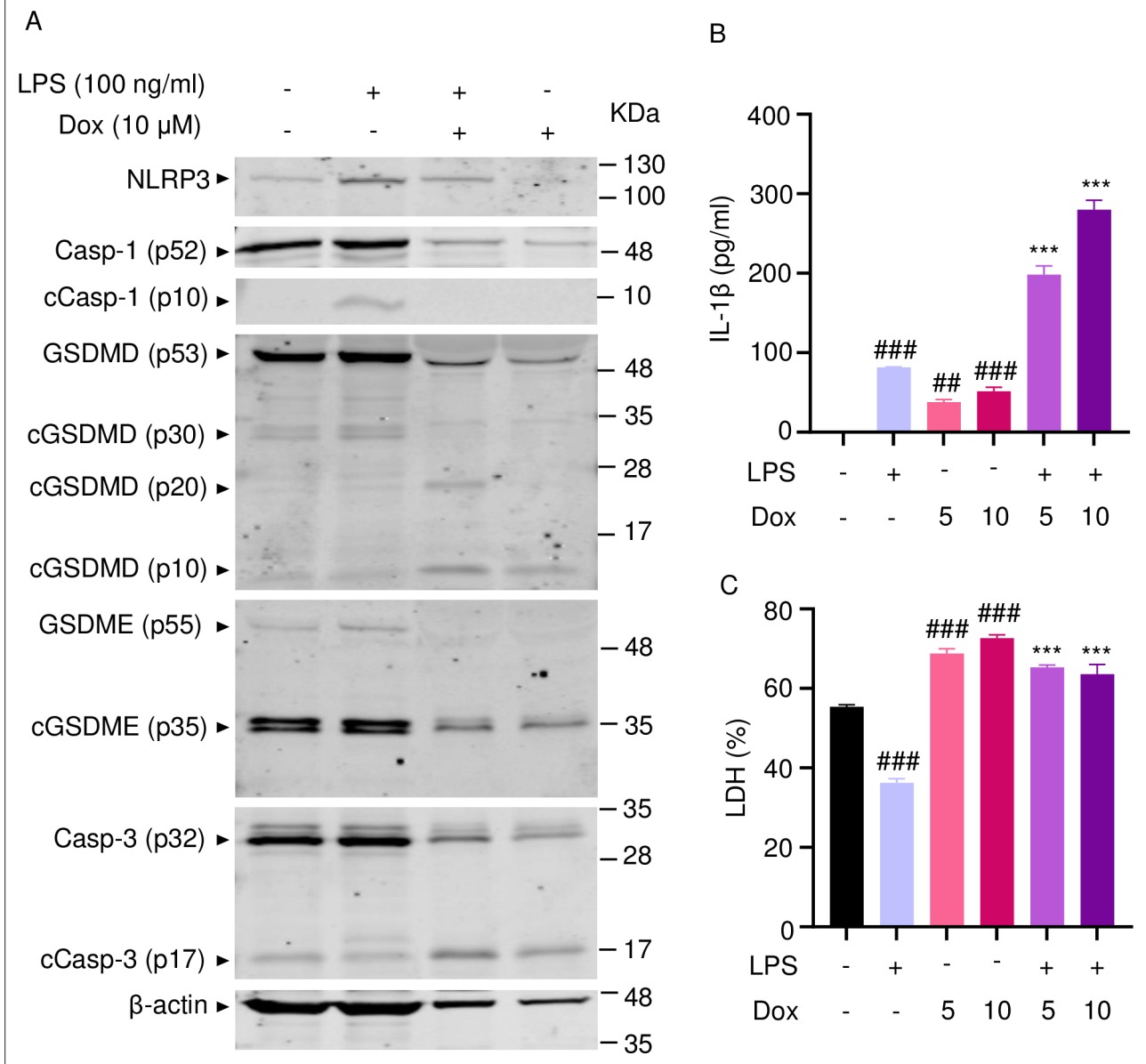

**Figure 5.** Doxorubicin activates inflammasome-dependent and -independent pathways in neutrophils. WT mouse bone marrow neutrophils were left untreated or primed with LPS for 3 hr, then treated with various doxorubicin concentrations for 16 hr. Whole cell lysates were analyzed by immunoblotting. Blots are representative of at least three independent experiments (**A**), IL-1β (**B**), and LDH (**C**) in the conditioned media were measured by enzyme linked immunosorbent assay (ELISA) and the cytotoxicity detection kit, respectively. Data are mean ± SEM from experimental triplicates and are representative of at least two independent experiments. ***p<0.001 vs. LPS; ##p<0.01, ###p<0.001 vs. untreated cultures. One-way ANOVA was used. cCasp, cleaved caspase; cGSDM, cleaved gasdermin; IL-1β, interleukin-1β; LDH, lactate dehydrogenase; LPS, lipopolysaccharide; Dox, doxorubicin.

The online version of this article includes the following source data and figure supplement(s) for figure 5:

**Source data 1.** Original file for the western blot analysis in *Figure 5A* (NLRP3).

**Source data 2.** Original file for the western blot analysis in *Figure 5A* (caspase-1/cleaved caspase-1).

**Source data 3.** Original file for the western blot analysis in *Figure 5A* (gasdermin D [GSDMD]/cleaved GSDMD).

**Source data 4.** Original file for the western blot analysis in *Figure 5A* (GSDME/cleaved GSDME).

**Source data 5.** Original file for the western blot analysis in *Figure 5A* (caspase-3/cleaved caspase-3).

**Source data 6.** Original file for the western blot analysis in *Figure 5A* (β-actin).

**Source data 7.** Original images for the western blot analysis (NLRP3, caspase-1/cleaved caspase-1, gasdermin D [GSDMD]/cleaved GSDMD, GSDME/cleaved GSDME, caspase-3/cleaved caspase-3, and β-actin) with highlighted bands and sample labels in *Figure 5A*.

*Figure 5 continued on next page*

*Figure 5 continued*

**Source data 8.** IL-1β and lactate dehydrogenase (LDH) analysis of bone marrow neutrophils supernatant in *Figure 5B and C*.

**Figure supplement 1.** Doxorubicin activates inflammasome-dependent and -independent pathways in neutrophils.

**Figure supplement 1—source data 1.** Western blot analysis of bone marrow neutrophils in *Figure 5—figure supplement 1*.

## Peripheral blood analysis

Mouse blood was collected by cardiac puncture in the EDTA-containing tubes. Complete blood counts were performed by the Washington University School of Medicine as previously described (*Wang et al., 2017*).

## Cell cultures

Murine primary BMMs were obtained by culturing mouse bone marrow cells in culture media containing a 1:10 dilution of supernatant from the fibroblastic cell line CMG 14-12 as a source of macrophage colony-stimulating factor, a mitogenic factor for BMMs, for 4–5 days in a 15 cm dish as previously described (*Takeshita et al., 2000*; *Wang et al., 2020*). Briefly, nonadherent cells were removed by vigorous washes with PBS, and adherent BMMs were detached with trypsin-EDTA and cultured in culture media containing a 1:10 dilution of CMG for various experiments. Murine primary neutrophils were isolated by collecting bone marrow cells and subsequently over a discontinuous Percoll (Sigma-Aldrich, MO, USA) gradient as described previously (*Sun et al., 2022*). Briefly, all bone marrow cells from femurs and tibias were washed by PBS and then resuspended in 2 ml PBS. Cell suspension was gently layered on top of gradient (72% Percoll, 64% Percoll, 52% Percoll) and centrifuged at 1545×$g$ for 30 min at room temperature. After carefully discarding the top two cell layers, the third layer containing neutrophils was transferred to a clean 15 ml tube. Cells were washed and counted, then plated at a density of $1\times10^5$ cells/well in 96-well plate or $5\times10^6$ cells/well in six-well plate for 1 hr followed by various experiments. For all in vitro experiments except otherwise specified, BMMs were plated at $2\times10^4$ cells per well on a 96-well plate or $10^6$ cells per well on a six-well plate overnight. Neutrophils were plated at $10^5$ cells per well on a 96-well plate or $5\times10^6$ cells per well on a six-well plate for 1 hr prior to treatment. BMMs and neutrophils were primed with 100 ng/ml LPS (Sigma-Aldrich, MO, USA) for 3 hr, then with different concentrations of doxorubicin (Sigma-Aldrich, MO, USA) as indicated for 16 hr. Conditioned media were collected for the analysis of IL-1β and LDH. Cell lysates were collected for protein expression analysis by western blot as described below.

## Western blot analysis

Cell extracts were prepared by lysing cells with RIPA buffer (50 mM Tris, 150 mM NaCl, 1 mM EDTA, 0.5% NaDOAc, 0.1% SDS, and 1.0% NP-40) plus phosphatase and protease inhibitors (GenDEPOT, TX, USA). Protein concentrations were determined by the Bio-Rad Laboratories method (Bio-Rad, CA, USA), and equal amounts of proteins were subjected to SDS-PAGE gels (12% or 15%) as previously described (*Wang et al., 2021*). Proteins were transferred onto nitrocellulose membranes and incubated with antibodies against GSDMD (1;1000; Abcam, MA, USA; ab219800; ab209845), GSDME (1;1000; Abcam, MA, USA; ab215191), caspase-1 (1;1 000; Abcam, MA, USA; ab179515), caspase-3 (1:1000; Cell Signaling Technologies, MA, USA; 9662S), NLRP3 (1:1000; AdipoGen, CA, USA; AG20B0014C), AIM2 (1:1000; Cell Signaling Technologies, MA, USA; 63660S) or β-actin (1:2000; Santa Cruz Biotechnology, TX, USA; SC47778) overnight at 4°C followed by incubation for 1 hr with secondary goat anti-mouse IRDye 800 (Li-COR Biosciences, NE, USA; 926-32210) or goat anti-rabbit Alexa Fluor 680 (Li-COR Biosciences, NE, USA; 926-68071) respectively. The results were visualized using the Odyssey infrared imaging system (LI-COR Biosciences, NE, USA).

## LDH assay and IL-1β ELISA

Cell death was assessed by the release of LDH in conditioned medium using LDH cytotoxicity detection kit (TaKaRa, CA, USA). IL-1β levels in conditioned media were measured by an ELISA kit (eBiosciences, NY, USA).

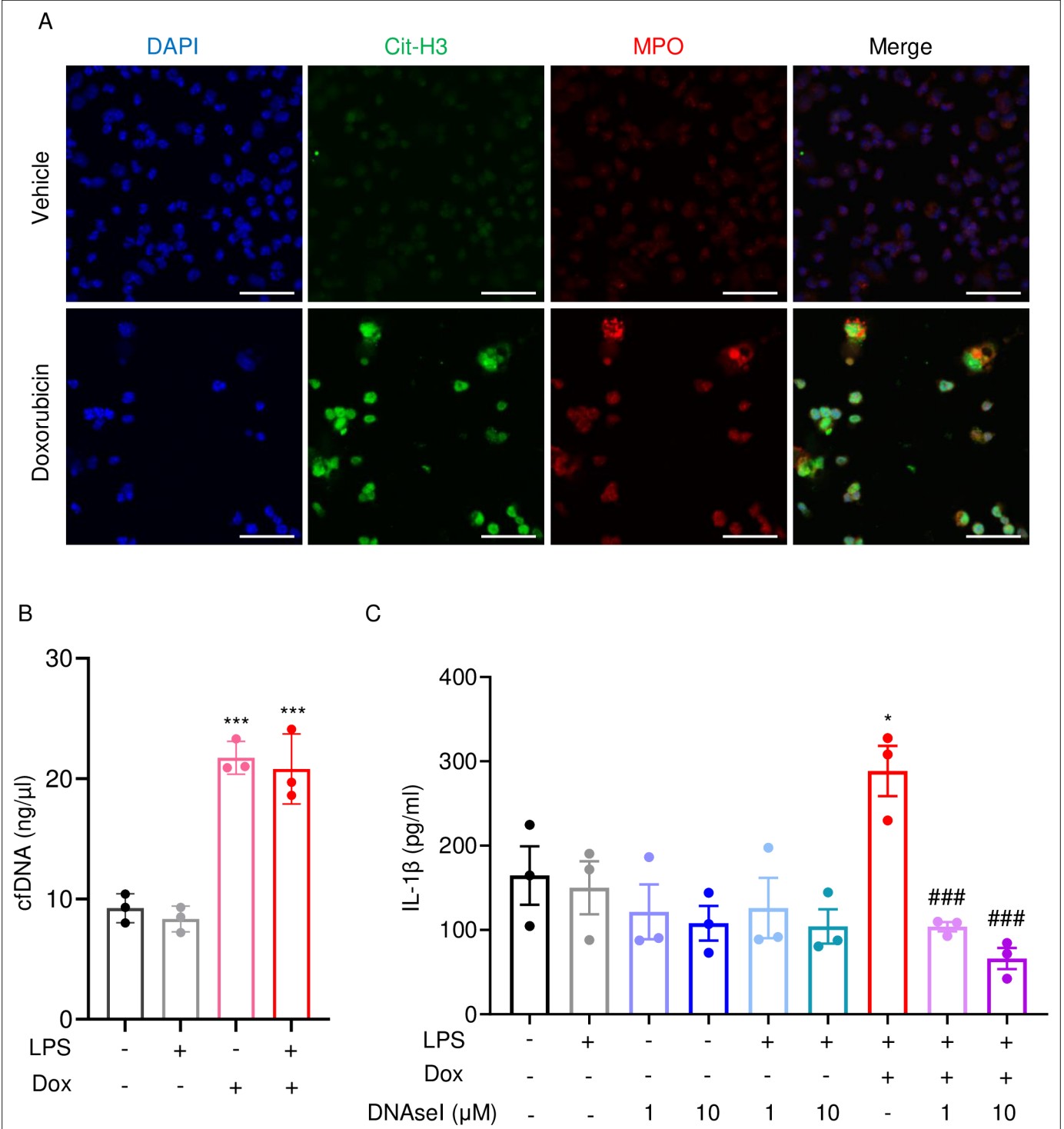

**Figure 6.** Doxorubicin causes NETosis in vitro. Wild-type (WT) mouse bone marrow neutrophils were left untreated or treated with 10 μM doxorubicin for 16 hr (**A**). Cit-H3 and MPO were analyzed by immunofluorescence. Scale bars: 50 μm. Images are representative of at least three independent experiments. Neutrophils were left untreated or primed with LPS for 3 hr, then treated with 10 μM doxorubicin for 16 hr. cfDNA in the conditioned medium was extracted and quantified (**B**). Neutrophils were left untreated or primed with LPS for 3 hr, then treated with 10 μM doxorubicin and/or DNase I for 16 hr. IL-1β in the conditioned media was measured by enzyme linked immunosorbent assay (ELISA) (**C**). Data are mean ± SEM from experimental triplicates and are representative of at least two independent experiments. *p<0.05; **p<0.01; ***p<0.001 vs. LPS; #p<0.05, ###p<0.001 vs. LPS+Dox. One-way ANOVA was used. Dox, doxorubicin; cfDNA, cell-free DNA; Cit-H3, citrullinated histone H3; MPO, myeloperoxidase.

The online version of this article includes the following source data for figure 6:

**Source data 1.** Cell-free DNA (cfDNA) and IL-1β analysis of bone marrow neutrophils supernatant in *Figure 6B–C*.

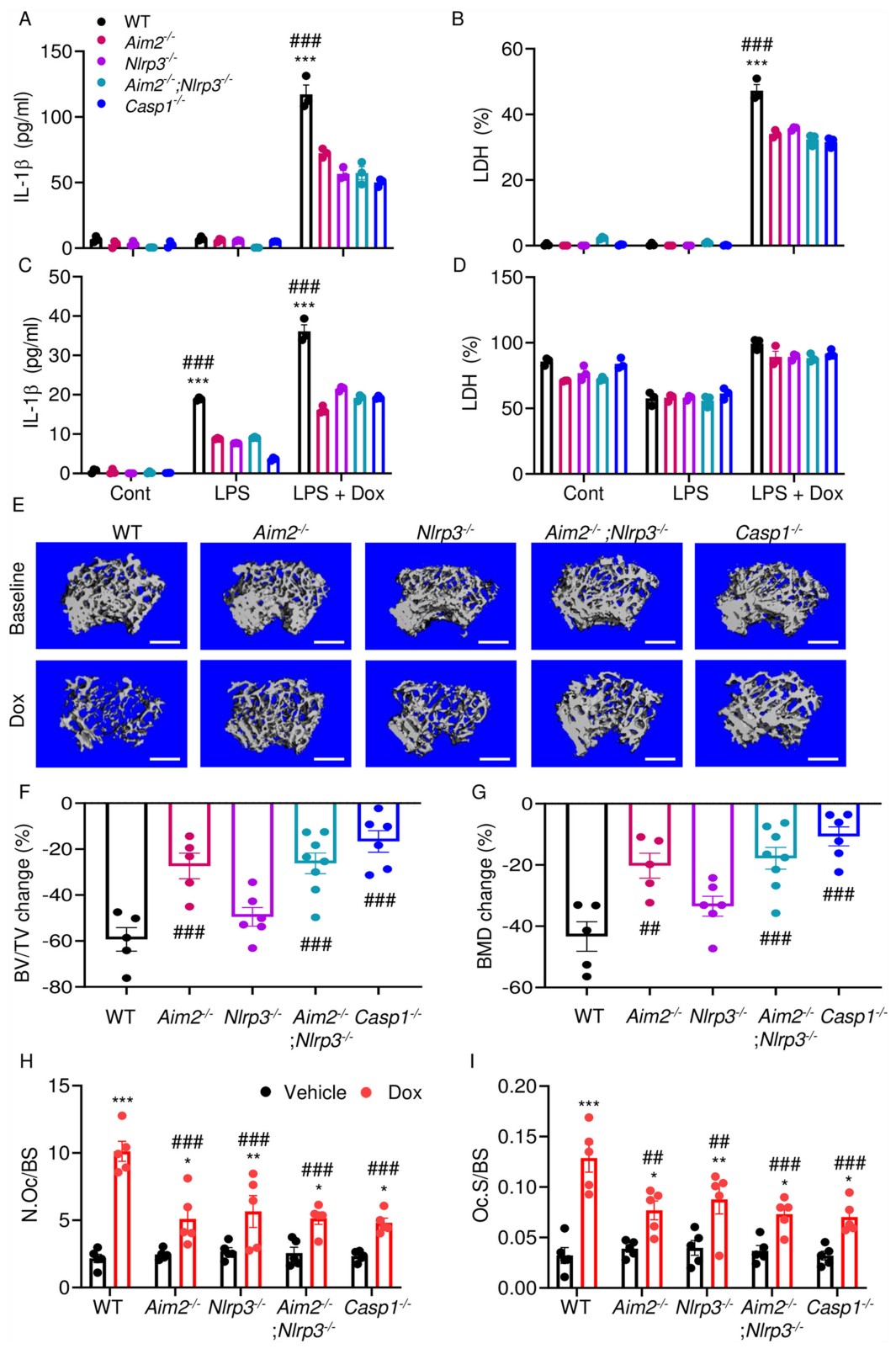

**Figure 7.** AIM2 and NLRP3 inflammasomes are involved in bone-damaging effects of doxorubicin. WT, *Aim2-/-*, *Nlrp3-/-*, *Aim2-/-;Nlrp3-/-* or *Casp1-/-* bone marrow-derived macrophages (BMMs) (**A, B**) and neutrophils (**C, D**) were left untreated or primed with LPS for 3 hr, then exposed or not to 10 µM doxorubicin for 16 hr. IL-1β (**A, C**) and LDH (**B, D**) in the conditioned media were measured by enzyme linked immunosorbent assay (ELISA) and the

*Figure 7 continued on next page*

*Figure 7 continued*

cytotoxicity detection kit, respectively. Femurs from male mice were analyzed by VivaCT before (baseline) and 4 weeks after a single intraperitoneal injection of 5 mg/kg doxorubicin (**E–G**). Cross sections of 3D reconstructions. Scale bars: 500 µm (**E**). BV/TV changes (**F**). BMD changes (**G**). Femurs harvested from different genotypes of male mice were analyzed 4 weeks after a single intraperitoneal injection of vehicle or doxorubicin (**H, I**). Specimens were stained for tartrate-resistant acidic phosphatase (TRAP) activity. N.Oc/BS (**H**). Oc.S/BS (**I**). Data are mean ± SEM from experimental triplicates and are representative of at least two independent experiments (**A–D**); n=5–8 mice/group (**E–I**). Data are mean ± SEM. *p<0.05; **p<0.01; ***p<0.001 vs. control, LPS, or vehicle; ##p<0.01, ###p<0.001 vs. other genotypes or WT treated with Dox. Two-way ANOVA (**A–D, H–I**) and one-way ANOVA (**F–G**) were used. AIM2, absent in melanoma 2; BMD bone mineral density; BV/TV, bone volume/total volume; casp1, caspase-1; Cont, control; Dox, doxorubicin; IL-1β, interleukin-1β; LDH, lactate dehydrogenase; LPS, lipopolysaccharide; N.Oc/BS, OC number/bone surface; Oc.S/BS; OC, osteoclast; WT, wild-type.

The online version of this article includes the following source data and figure supplement(s) for figure 7:

**Source data 1.** IL-1β and lactate dehydrogenase (LDH) analysis of different genotypes of bone marrow-derived macrophages (BMMs) and bone marrow neutrophils supernatant in *Figure 7A–D*.

**Source data 2.** Micro-computed tomography (µCT) analysis of different genotypes of mice in *Figure 7F and G*.

**Source data 3.** Tartrate-resistant acidic phosphatase (TRAP) staining analysis of different genotypes of mice in *Figure 7H and I*.

**Figure supplement 1.** AIM2 and NLRP3 inflammasomes are involved in bone-damaging effects of doxorubicin in male mice.

**Figure supplement 1—source data 1.** Micro-computed tomography (µCT) analysis of different genotypes of male mice in *Figure 7—figure supplement 1B*.

**Figure supplement 2.** AIM2 and NLRP3 inflammasomes are involved in bone-damaging effects of doxorubicin in female mice.

**Figure supplement 2—source data 1.** Micro-computed tomography (µCT) analysis of different genotypes of female mice in *Figure 7—figure supplement 2B–D*.

## ASC specks assay

ASC-citrine-WT BMMs were plated at $10^4$ cells per well on a 16-well glass plate overnight. Cells were primed with LPS for 3 hr followed by 15 µM nigericin (AdipoGen, CA, USA) for 30 min or the indicated doxorubicin concentrations for 16 hr. Cells were washed with PBS, fixed with 4% paraformaldehyde buffer for 10 min at room temperature, then counterstained with Fluoro-gel II containing DAPI (Fluoro-Gel, Fisher Scientific Intl INC, PA, USA). ASC-citrine photographs were taken using ZEISS microscopy (Carl ZEISS Industrial Metrology, MN, USA). Quantification of ASC specks was carried out using ImageJ.

## Immunofluorescence

Isolated neutrophils were plated at $10^5$ cells per well on a 16-well glass plate for 1 hr. Cells were primed with LPS for 3 hr, treated with doxorubicin for 16 hr, washed with PBS, and fixed with 4% paraformaldehyde buffer for 10 min at room temperature. Cells were permeabilized with 0.2% Triton in PBS for 20 min, blocked with 0.2% Triton and 1% BSA in PBS for 30 min, and were incubated with Cit-H3 antibody (1:1000; Abcam, MA, USA; ab5103) and MPO (1;1000; Abcam, MA, USA; ab90810) overnight at 4°C in blocking buffer, followed by incubation with secondary antibody (Alexa Fluor 594, 1:2000; Life Technologies, CA, USA; A11020; A27034) for 30 min. Cells were counterstained with Fluoro-gel II containing DAPI (Fluoro-Gel, Fisher Scientific Intl INC, PA, USA). Immunostaining images were taken using a Leica inverted microscope with a TCS SPEII confocal module and processed using LAS X software (Leica Microsystems Inc, IL, USA).

## cfDNA assay

cfDNA in the conditioned cell culture medium was extracted using NucleoSpin Gel and PCR Clean-up kit (Takara, Duren, Germany), and quantified with Nanodrop (Thermo Fisher Scientific, MA, USA). cfDNA in the serum was purified and measured using Qubit by Washington University School of Medicine Hope Center DNA/RNA purification Core or using SYBR Green (Thermo Fisher Scientific, MA, USA) as previously described (*Goldshtein et al., 2009*; *Villalba-Campos et al., 2016*).

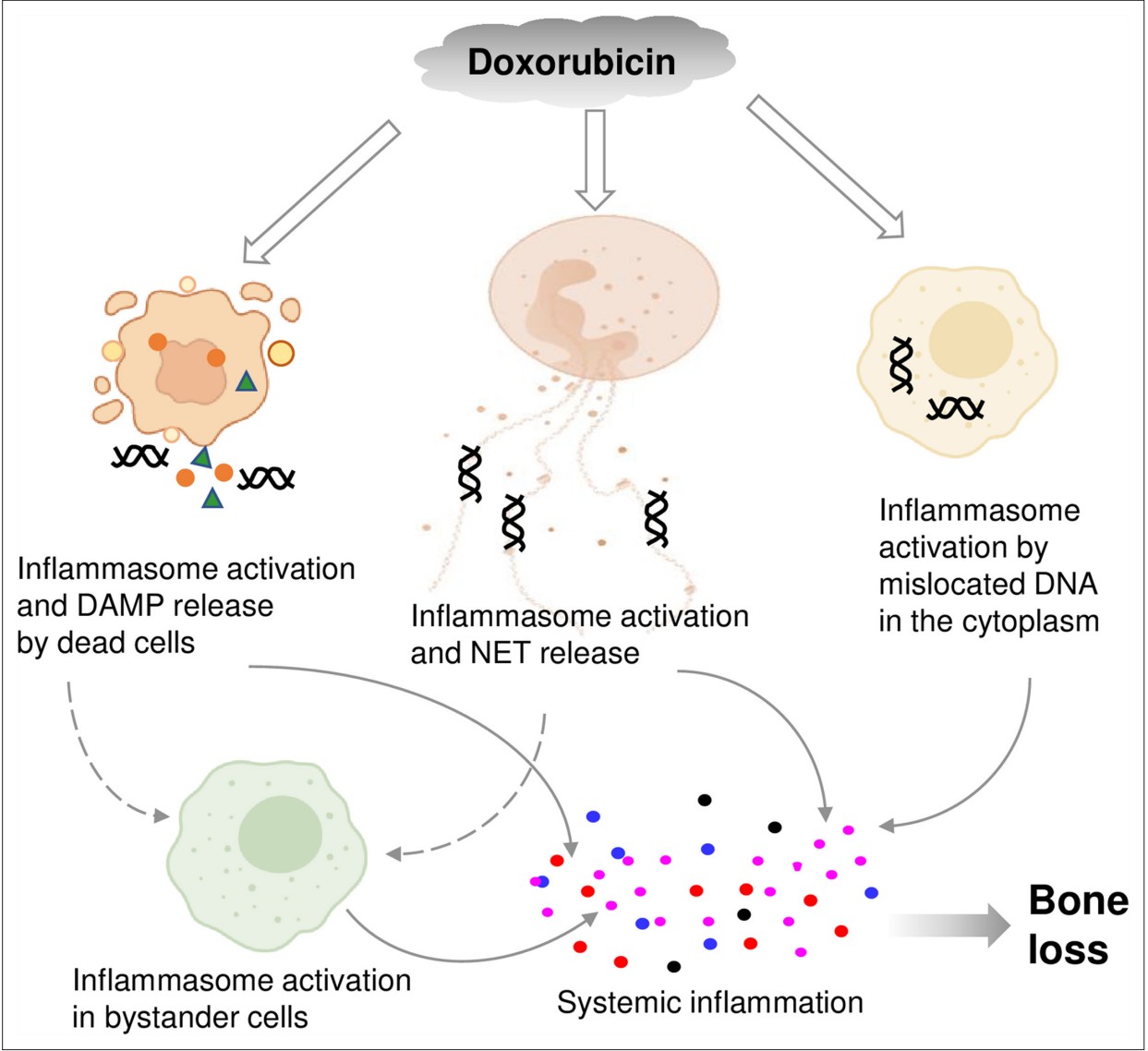

**Figure 8.** Graphical abstract. Double line arrows: direct effects of doxorubicin on target cells. Solid line arrows: direct contribution to systemic inflammation. Broken line arrows: indirect contribution to systemic inflammation.

### ATP assay

ATP levels in conditioned media and serum were measured by RealTime-Glo Extracellular ATP Assay kit (Promega, Madison, WI, USA).

### Statistical analysis

Statistical analysis was performed using the Student's t-test, one-way ANOVA with Tukey's multiple comparisons test, or two-way ANOVA with Tukey's multiple comparisons test, Dunnett's multiple comparisons test, or Sidak's multiple comparisons test using the GraphPad Prism 9.0 software. Values are expressed as mean ± SEM. *$p < 0.05$ was considered statistically significant.

### Acknowledgements

We want to thank Dr. Deborah J Veis for reading this manuscript.

# Additional information

## Competing interests

Canxin Xu: Employee of Aclaris Therapeutics, Inc. Gabriel Mbalaviele: Holds stocks of Aclaris Therapeutics, Inc. The other authors declare that no competing interests exist.

## Funding

| Funder | Grant reference number | Author |
|---|---|---|
| National Institute of Arthritis and Musculoskeletal and Skin Diseases | R01-AR072623 | Yousef Abu-Amer |
| National Institute of Arthritis and Musculoskeletal and Skin Diseases | R01-AR082192 | Yousef Abu-Amer |
| Shriners Hospitals for Children | 85109 | Yousef Abu-Amer |
| National Institute of Arthritis and Musculoskeletal and Skin Diseases | P30 AR074992 | Yousef Abu-Amer |
| National Institute of Arthritis and Musculoskeletal and Skin Diseases | R01-AR076758 | Gabriel Mbalaviele |
| National Institute of Allergy and Infectious Diseases | R01-AI161022 | Gabriel Mbalaviele |
| National Institute on Aging | R01 AG077732 | Gabriel Mbalaviele |

The funders had no role in study design, data collection and interpretation, or the decision to submit the work for publication.

## Author contributions

Chun Wang, Data curation, Formal analysis, Investigation, Methodology, Writing – original draft, Writing – review and editing; Khushpreet Kaur, Data curation, Formal analysis, Investigation, Methodology, Writing – review and editing; Canxin Xu, Data curation, Formal analysis, Methodology; Yousef Abu-Amer, Resources, Funding acquisition, Writing – review and editing; Gabriel Mbalaviele, Conceptualization, Formal analysis, Funding acquisition, Investigation, Writing – original draft, Project administration, Writing – review and editing

## Author ORCIDs

Yousef Abu-Amer (iD) http://orcid.org/0000-0002-5890-5086
Gabriel Mbalaviele (iD) https://orcid.org/0000-0003-4660-0952

Reviewer #1 (Public Review): https://doi.org/10.7554/eLife.92885.4.sa1
Reviewer #2 (Public Review): https://doi.org/10.7554/eLife.92885.4.sa2
Author response https://doi.org/10.7554/eLife.92885.4.sa3

# Additional files

## Supplementary files

- MDAR checklist

## Data availability

All data generated or analysed during this study are included in the manuscript and supporting files; source data files have been provided for all figures.

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
