## [Editor Report · eLife assessment]

This **useful** study, which systematically addresses off-target effects of a commonly used chemotherapy drug on bone and bone marrow cells and which therefore is of potential interest to a broad readership, presents evidence that reducing systemic inflammation induced by doxorubicin limits bone loss to some extent. The demonstration of the effect of systemic inflammation on bone loss is **convincing**. Building on prior work, this study sets the scene for additional genetic and pharmacologic experiments as well as future analyses of the bone phenotypes, which should speak to the mechanisms involved in doxorubicin-induced bone loss – which are not addressed in the current study – and which may substantiate the clinical relevance of targeting inflammation in order to limit the negative impact of chemotherapies on bone quality.

---

## [Referee Report · Reviewer #1 (Public Review)]

Summary:

Doxorubincin has long been known to cause bone loss by increasing osteoclast and suppressing osteoblast activities. The study by Wang et al. reports a comprehensive investigation into the off-target effects of doxorubicin on bone tissues and potential mechanisms.. They used a tumor-free model with wild type mice and found that even a single dose of doxorubicin has a major influence by increasing leukopenia and DAMPs and inflammasomes in macrophages and neutrophils, and inflammation-related cell death (pyroptosis and NETosis). The gene knockout study shows that AIM2 and NLRP3 are the major contributors to bone loss. Overall, the study confirmed previous findings regarding the impact of doxorubicin on tissue inflammation and expands the research further into bone tissue. The presented data presented are consistent; however, a major question remains regarding whether doxorubicin drives inflammation and its related events. Most in vitro study showed that the effect of doxorubincin cannot be demonstrated without LPS priming. This observation raises the question of whether doxorubincin itself could activate the inflammasome and the related events. In vivo study, on the other hand, suggested that it doesn't require LPS. The inconsistency here was not explained further. Moreover, a tumor-free mouse model was used for the study; however, immune responses in tumor bearing models would likely be distinct from tumor-free ones. The justification for using tumor-free models is not well-established.

Strengths:

The paper includes a comprehensive study that shows the effects of doxorubincin on cytokine levels in serum, release of DAMPs and NETosis, and leukopenia using both in vivo and in vitro models. Bone marrow cells, macrophages and neutrophils were isolated from the bone marrow, and the levels of cytokines in serum were also determined.

They employed multiple knockout models with deficiency in Aim 2, Nlirp3, and double deficiencies to dissect the functional involvement of these two inflammasomes.

The experiments in general are well designed. The paper is also logically written, and figures were clearly labeled.

Weaknesses:

Most of the data presented are correlative, and there is not much effort to dissect the underlying molecular mechanism.

It is not entirely clear why a tumor free model is chosen to study immune responses, as immune responses can differ significantly with or without tumor-bearing.

Immune responses in isolated macrophages, neutrophils and bone marrow cells require priming with LPS, while such responses are not observed in vivo. There is no explanation for these differences.

The band intensities on Western blots in Fig. 4 and Fig. 5 are not quantified, and the numbers of repeats are also not provided.

Many abbreviations are used throughout the text, and some of the full names are not provided.

Fig. 5B needs a label on X axis.

---

## [Referee Report · Reviewer #2 (Public Review)]

Summary:

Wang and collaborators have evaluated the impact of inflammation on bone loss induced by Doxorubicin, which is commonly used in chemotherapy to treat various cancers. In mice, they show that a single injection of Doxorubicin induces systemic inflammation, leukopenia, and a significant bone loss associated with increased bone-resorbing osteoclast numbers. In vitro, the authors show that Doxorubicin activates the AIM2 and NLRP3 inflammasomes in macrophages and neutrophils. Importantly, they show that the full knockouts (germline deletions) of AIM2 (Aim2-/-) and NLRP3 (Nlrp3-/-) and Caspase 1 (Casp1-/-) limit (but do not completely abolish) bone loss induced 4 weeks after a single injection of Doxorubicin in mice. From these results, they conclude that Doxorubicin activates inflammasomes to cause inflammation-associated bone loss.

Strength:

This manuscript provides functional experiments demonstrating that NRLP3 and/or AIM2 loss-of-functions (and thus the systemic impairment of the inflammatory response) prevent bone-loss induced by Doxorubicin in mice.

Weaknesses:

Numerous studies have reported that Doxorubicin induces systemic inflammation and activates the inflammasome in myeloid cells and various other cell types. It is also known that systemic inflammation and Doxorubicin treatment lead to bone loss. Hence, the key conclusions drawn from this work have been known already or were very much expected. Therefore, the novelty appears somewhat limited. One important limitation is the lack of experiments that could determine which cell lineages are involved in bone loss induced by Doxorubicin in vivo, while the tools to do so exist. The characterization of the bone phenotype is incomplete, and unfortunately does not tell us whether the inflammasome is activated in some of the cell lineages present in bones in vivo. Another limitation is that the relative importance of the inflammasomes compared to cell senescence and autophagy, which are also induced by Doxorubicin, has not been evaluated. Hence the main molecular mechanisms responsible for bone loss induced by Doxorubicin in vivo remains unknown. Lastly, it would have been interesting, on a more clinical point of view, to compare the few relevant treatments that could limit the deleterious effect of Doxorubicin on bone loss while preserving the toxicity on tumor cells.

---

## [Author Response]

The following is the authors’ response to the previous reviews.

**Reviewing Editor**
We thank you for clarifying several of the questions raised by the reviewers. Since the study has otherwise largely stayed unchanged, we will leave the eLife assessment as “before”:

We respectfully disagree because we addressed all concerns raised by the two reviewers except one (below), which was not satisfactorily answered according to reviewer 1; it has now been addressed (new S3 Fig).

**Reviewer #1 (Recommendations For The Authors):**
The authors addressed most of my previous comments. However, there is one important point that was not satisfactorily addressed "The band intensities on Western blots in Fig. 4 and Fig. 5 are not quantified, and the numbers of repeats are also not provided" The response that "It is not straightforward to quantify and describe the intensity of the bands of these numerous with different fate outcomes." In the revision, they mentioned at least three repeats were performed. If so, it's not entirely clear why they couldn't quantify the western blots results. Including quantitative data will strengthen the rigor of the findings.

Quantitative data from Fig. 4 and Fig. 5 are now provided as S3 Fig and described in the manuscript (lines 170-175; 184-188).